# A Panoramic View on Grapevine Trunk Diseases Threats: Case of Eutypa Dieback, Botryosphaeria Dieback, and Esca Disease

**DOI:** 10.3390/jof8060595

**Published:** 2022-06-01

**Authors:** Jihane Kenfaoui, Nabil Radouane, Mohammed Mennani, Abdessalem Tahiri, Lahsen El Ghadraoui, Zineb Belabess, Florence Fontaine, Hajar El Hamss, Said Amiri, Rachid Lahlali, Essaid Ait Barka

**Affiliations:** 1Phytopathology Unit, Department of Plant Protection, Ecole Nationale d’Agriculture de Meknès, Km10, Rte Haj Kaddour, BP S/40, Meknes 50001, Morocco; jihane.kenfaoui@usmba.ac.ma (J.K.); nabil.radouane@usmba.ac.ma (N.R.); mmennani@enameknes.ac.ma (M.M.); atahiri@enameknes.ac.ma (A.T.); hajar.elhamss@gmail.com (H.E.H.); samiri@enameknes.ac.ma (S.A.); 2Laboratory of Functional Ecology and Environmental Engineering, Sidi Mohamed Ben Abdellah University, P.O. Box 2202, Route d’Imouzzer, Fez 30500, Morocco; lahsen.elghadraoui@usmba.ac.ma; 3Plant Protection Laboratory, Regional Center of Agricultural Research of Oujda, National Institute of Agricultural Research, Avenue Mohamed VI, BP428 Oujda, Oujda 60000, Morocco; zineb.belabess@inra.ma; 4Unité de Recherche Résistance Induite et Bio-Protection des Plantes-EA 4707 USC INRAE 1488, Université de Reims Champagne-Ardenne, 51100 Reims, France; florence.fontaine@univ-reims.fr

**Keywords:** vine, phytosanitary problems, fungus, biological control

## Abstract

Grapevine trunk diseases (GTD) are currently one of the most devastating and challenging diseases in viticulture, leading to considerable yield losses and a remarkable decline in grapevine quality. The identification of the causal agents is the cornerstone of an efficient approach to fighting against fungal diseases in a sustainable, non-chemical manner. This review attempts to describe and expose the symptoms of each pathology related to GTD, the modes of transmission, and the harmfulness of recently reported agents. Special attention was given to new diagnostic tests and technologies, grapevine defense mechanisms, molecular mechanisms of endophytes fungal colonization, and management strategies used to control these threats. The present extended review is, therefore, an updated state-of-the-art report on the progress in the management of vineyards.

## 1. Introduction

Viticulture is an agricultural practice that dates back over 7000 years [1]. It is practiced on all continents except Antarctica. Overall, the world area under grapevines in 2020 is estimated to be 7.3 million hectares (mha), which corresponds to the total surface area planted with grapevines for all purposes (wine and juices, table grapes, and raisins), including young grapevines not yet in production [2]. Originally from the Middle East and the Mediterranean Basin, the history of viticulture is intimately linked to that of Morocco, as the first vines were planted by the Phoenicians and Roman colonists [3]. This culture took off at the beginning of the twentieth century with the protectorate and continues to climb the ladder of the best vineyards in the world. Thus, Morocco is the second-largest producer of wine in the Arab world, and also one of the cradles of the last wild vines.

In Morocco, the “Land of Vines”, historically known as the wine paradise of the Mediterranean, plays a key socio-economic role. It occupies an area of more than 50,000 hectares with an annual production of about 452,000 tons, divided into 346,656.26 tons of table grape, which are concentrated in the regions of Doukkala, Tansies-EL Haouz, Benslimane, Essaouira, Khémisset, and Rabat-Salé, while 104,501.25 tons of wine-grape are mainly concentrated in the regions of El-Hajeb, Khémisset, Meknes, Gharb and Moulouya [4]. On the social level, the vitivinicultural sector provides almost 7.2 million working days per year or 30,500 permanent jobs. Exports are a significant source of foreign currency of about USD 10952902.00. However, this wine-producing activity is unfortunately threatened by a series of phytosanitary problems, which could jeopardize this precious Moroccan floristic heritage preserved for centuries [5].

The different vine varieties belong to a common systematic trunk: a dicotyledonous angiosperm plant of the *Vitaceae* family. The genus *Vitis* comprises two subgenera, *Muscadinia* and *Vitis*, with distinct anatomical, morphological and cytological characteristics. The subgenus *Vitis* includes about sixty species of which *Vitis vinifera* L. is the most commercialized in the world [6].

One of the major difficulties of grapevine cultivation is its susceptibility to a wide range of microorganisms, such as viruses, bacteria, nematodes, and fungi [7], causing varying degrees of damage in vineyards. The most common ones are downy mildew (*Plasmopara viticola*), powdery mildew (Erysiphe necator), anthracnose (*Elsinoe ampelina*), greenaria bitter rot (*Greeneria uvicola*) and black rot (*Guignardia bidwellii*). The damage can range from simple reactions of the plant to devastating effects [8]. In recent years, there has been a resurgence of emerging diseases affecting the vine. These biotic stresses are commonly known as grapevine trunk diseases (GTD). In Morocco, these pathologies are taking the forefront of winegrowers’ concerns, as they cause considerable damage in vineyards not only quantitatively but also qualitatively [9]. As a result, it is critical to properly identify the diseases present in the vineyard to prevent serious infections and loss of yield or quality. To address this issue, winegrowers turned to the use of sodium arsenite, which allowed them to contain the expression of these diseases and reduce field losses. Since the ban of arsenite sodium in 2003 in Europe due to its adverse effects on human health and the environment, no effective alternative curative replacement has been proposed to the profession [10]. Therefore, the incidence of GTD symptoms has been observed, especially in younger vineyards [11].

Currently, the increasing incidence of GTD is a serious threat to the economic viability of the vineyards in some wine-growing regions, including Moroccan vineyards. The type and severity of diseases in the wooded part of the vineyard vary according to the prevailing climate, cultivars, and fungal species. The etiology of these pathologies is very complex since numerous biotic, abiotic, and eco-physiological factors associated with cultural practices could explain the vineyard decline [12]. The incubation time required for the expression of GTD in the field complicates the assessment of suitable preventive solutions under both controlled and natural conditions [8].

In the lack of precise information concerning GTD, certain confusions relating mainly to the typical symptoms among these diseases, the biology of the causal agents, and the means of their transmission, can hinder the protection efficiency of the vineyard. To ensure holistic control and limit the impact of pathogens involved in GTD, it is essential to better understand the conditions and factors associated with their emergence as well as the biological and molecular mechanisms involved in the infection processes. The search for innovative methods to control fungal diseases is now a necessity.

Despite the importance of the vine in some regions in Morocco and worldwide, the production of grapes is not always guaranteed. Hence, few studies are carried out on GTDs in Morocco. The present review is an update of scientific knowledge on GTD, by targeting these fungal diseases in their entirety and diversity. We will also attempt to address the symptoms related to each pathology, the mode of transmission of the causal agents, and the harmfulness of recently reported pathogens. In addition, the new microbiological and molecular tests and technologies that are used to diagnose these fungal diseases, the grapevine defense mechanisms, the molecular mechanisms of endophytes fungal colonization, and reports on the latest management techniques used in the GTD control will be discussed.

## 2. The Complexity of Grapevine Trunk Diseases

The grapevine can be subjected to many fungal diseases, and GTDs are amongst the most common ones. Pascoe defined grapevine trunk diseases as, “all those diseases in which the pathogen is primarily located in the trunk (or cordons) of the grapevine and in which the main symptom is a slow decline as a result of interruption of xylem conductivity, and/or toxin production”. These diseases are due to the infection of the xylem tissues of mature wood by a pathogen and are, thus, permanent, deep-seated, difficult to diagnose, and difficult to target with standard treatments [13]. GTDs are recognized as the main identified cause of vine decline because they are detrimental to the sustainability of the viticultural heritage. The causal agents of these diseases are responsible for the death of the vine in the long term, requiring the renewal of plants that can reach more than 10% [14].

Because GTDs are more complex than other grapevine diseases such as powdery and downy mildew, managing them presents a dilemma for winegrowers, nurserymen, technicians, and scientists. One of the fascinating and controversial elements of GTD in field trials is their undefined latency period (asymptomatic phase) [15]. Symptoms may appear in year n and not in year n + 1 in the same vine due to environmental, climatic, and cultural factors [16,17], leading to an underestimation of the true incidence in the vineyard in any given year. During the many processes of plant production (hydration, cold storage, grafting, callusing, etc.), infected asymptomatic cuttings might cross-contaminate GTD pathogens), making latent infections harmful in the propagation process. If these infections are not managed, an unnoticed spread of diseased plants may occur, first in the nursery and subsequently throughout the vineyard [18].

GTDs, commonly known as decline diseases, include several disorders occurring under different symptomatologies. Studies of these diseases, conducted in several wine-growing areas worldwide, have shown the enormous number of fungal genera and species related to the diseases which often vary based on climate and geographical areas [19]. These investigations have also shown that their life cycles are inextricably linked to specific viticultural techniques, particularly dormant pruning. More specifically, pruning wounds are the main entry point for GTD pathogens into vines. GTD pathogens can infect wounds for up to four months, depending on the pathogen [20]. Because multiple viruses commonly associated with distinct GTDs can infect the wood of diseased vines at the same time, the internal and outward symptoms can intersect. Furthermore, even though neither cultivar nor species in the genus *Vitis* have been identified to display complete resistance to GTD, grapevine cultivars differ in their sensitivity to the development of foliar symptoms in specific GTDs [21].

Esca disease, Eutypa dieback, and Botryosphaeria dieback are the leading players of these diseases [22], which are associated with the presence of different fungi able to degrade woody tissues. These pathogenic fungi attack the vascular tissues, contaminating the perennial organs of the vine, thus causing the dieback of the plant. It is important to emphasize that the diversity of causal agents, the complexity of their biology, as well as the perceptible lack of available tools and control methods, are enough to complicate the linear association that occurs in other plant diseases. External symptoms are ineffective for diagnosing GTD efficiently and precisely. Traditional, molecular, and serological diagnostic approaches may lead to the identification of a group of related pathogens with varying virulence levels according to species or even strains, and that may have different sensitivity to a certain treatment. In this context, the lack of valid and simple control methods has hindered the feasibility of effective GTD control, as previously stated. All those elements combined, constitute a serious obstacle to developing the wine industry in Morocco and elsewhere [12].

### 2.1. Eutypa Dieback

An accurate description of Eutypa dieback or eutypiosis was provided in 1823. The nature of this disease was not entirely clarified until the end of the 19th century. Referred to as folletage by Pierre Viala (1859–1936) in 1887, its fungal origin was recognized by Luis Ravaz (1863–1937) in 1898 [23,24]. Eutypa dieback (formerly Eutypa dead arm) has been recognized for damage in various locations since the 1970s, including Australia, France, and California [25]. The first isolation of the pathogen, *Eutypa lata*, was carried out in 1900 from the apricot tree in Scotland. Subsequently, the fungus was first identified on the grapevine in 1978 [26]. At that time, it was called *Eutypa armeniacae*, but it was given the official name *E. lata* in 1984 [27].

Eutypa dieback is recorded in 88 species of dicotyledonous perennial woody plants, grouped in 52 genera and 27 families [28]. It has been observed on several plants species including lemon [29], apple [30], peach, almond [31], pistachio, apricot, blackcurrant, cherry, hazelnut, olive [32], tamarisk, and vine. The disease is present in almost all the vineyards of the world including Africa (South Africa [33], Algeria [34], and Morocco [35]), America (Mexico [36], Venezuela [37], Brazil [38], Canada [39], United States [40,41], and Uruguay [42]), Europe (Austria [43], France [44], Germany [45], Greece [46], Italy [47], Romania [48], Switzerland [49], Bulgaria [50], Croatia [51], Spain [52], Hungary [53], Portugal [54], Serbia [55], Slovakia [56]), Oceania (Australia [40]), New Zealand [57], and Sweden [49]), Asia (China [58], Iran [59], Jordan [60], Lebanon [61], Syria ([62], and Turkey [63]). The disease is absent in semi-desert areas (<250 mm rainfall per year) [31] (Figure 1).

This disease is caused by the ascomycete fungus *E. lata*. Two forms of the fungus are distinguished including asexual (or imperfect form) and sexual (or perfect form) [64]. The asexual form, called *Libertella blepharis*, appears on the surface of infected wood when humidity is high. It is characterized by the presence of subglobular pycnidia of 200 to 300 µm. These blackish and globose pycnidia emit a yellowish mucilaginous cord which contains a multitude of filiform conidia. Characteristic and arcuate stylospores (pycnidiospores), measuring 18–25 µm, are embedded within these conidia. On the wood, these form usually in areas where the perithecia appear later [47]. Those stylospores do not play a role in the spread of the disease [47] due to their difficulty to germinate and develop mycelium.

On the other hand, the sexual form *E. lata* is a highly polyphagous ubiquitous fungus living in both saprophytic and parasitic states, which ensures the spread of the disease. It develops under the bark (deadwood) and ensures its survival by the mycelium and the perithecia. The perithecia are sexual reproduction organs, which remain fertile up to 5 years after infection in the form of a blackish stroma with a bumpy appearance [65]. This contains the fruiting bodies or perithecia, which can only develop in regions where the annual rainfall exceeds 350 mm and constitute the source of infection [65].

Two other species are associated with this disease. These include *E. leptoplaca* in California [66] and *E. maura* in Jordan [67]. The pathogenicity of the first species is shown by the appearance of detached twigs or shoots. The fungus causes necrosis similar to those obtained with E. lata. However, no experimentation was carried out to determine whether it was responsible for the stunting observed in the vineyard. Regarding *E. maura*, no pathogenicity tests were performed. Another fungus, known as *Eutypella vitis*, is also found in Michigan on a grapevine showing eutypiosis symptoms [68]. Pathogenicity tests show that *E. vitis* can cause less severe symptoms on the herbaceous part and in the wood than those induced by the *E. lata* isolates [64].

In the spring, and due to the action of rain, the perithecia containing the asci burst and release the ascospores. The spread of ascospores is ensured by the wind over very large distances. Size wounds and frost injuries are the targets of ascospores that germinate in the xylem vessels. Thereafter, the fungus progresses slowly through the wood. It gradually colonizes wood vessels and adjacent tissues, developing sectorial necrosis. The plant variably resists the progression of mycelium, depending on the sensitivity of the grape variety [12].

*E. lata* produces toxins and virulence factors in the host plant. Spore entry sites are wood injuries, mainly pruning wounds but also injuries caused by frost, hail, and mechanical harvesting. The ascospores of *E. lata* can easily enter the wood vessels thanks to their small size and sticky surface, which ensure a high adhesion capacity [69]. During the fungal attack, the degradation of lignin, cellulose, and hemicellulose leads to the formation of cavities in the walls and thus, makes the wood brittle [70]. In vitro studies highlighted the strong lysis activity of plant cells (leaves or cell suspension) by fungal filtrate proteins, including hydrolytic enzymes such as chitinases, β-1.3-glucanases, glycosidases, and xylanases [71]. Some studies also revealed the presence of toxic molecules secreted by the fungus under in vitro conditions, including acetylene and heterocyclic compounds. The main one is Eutypin. It is composed of an aromatic part from the biosynthesis pathway of shikimic acid and a side chain from mevalonic acid [72,73]. This acid causes a variety of damage to plant cells. Once it penetrates the plant cells by passive diffusion, it is inserted into membranes due to its lipophilic property. This causes cytoplasm acidification, a decrease in leucine transport, hypertrophy of chloroplasts, an expansion of the thylakoids, plasmalemma retraction, cytoplasm lysis, and vesiculation endomembranes [74]. This secondary metabolite is absent in healthy vines but was found in the sap, leaves, stems, and inflorescences of infected plants. Therefore, *E. lata* has an important arsenal for infecting, occupying, and degrading vine wood, which made it difficult to control this pathogen [75].

Eutypa dieback can develop in the wood over years without any symptoms. Then, the presence of the disease will be revealed only after the appearance of the first symptoms characterized by leaf damping off, which are attributed to the production of fungal toxins. This partly explains the difficulty of establishing a correct diagnosis in the early stages of the infection [76]. Eutypiosis symptoms are most visible in spring [12]. They are characterized by leaf chlorosis and drying, accompanied by dwarfing of the herbaceous organs, following shortening of the internodes, and modification of the leaf structure. The tense and the chlorotic aspect of the leaves lead to a photosynthetic decline giving the diseased arms a bushy appearance with greyish or blackish areas. The latter correspond to the perithecia that materialize the presence of the fungus [77] (Figure 2a). In the wood, these symptoms appear as hard, brown, and sectorial necrosis with a dry appearance that develops from the wound area and extends into the rest of the wood [12] (Figure 2b). These necroses are due to enzymes that degrade the plant cell wall [65]. All these damages affect the production and longevity of the vineyard, leading to substantial economic losses.

### 2.2. Esca Disease

Esca disease is one of the most destructive diseases affecting grapevine crops in the world [78]. The first references to Esca symptoms are mentioned in several ancient Greek and Latin works. However, more precise descriptions date back to medieval times [79]. This pathology has been described in its apoplectic form, and its origin was long undetermined and attributed to a physiological disorder that was referred to as drying-out [80,81]. This form was only assigned to a fungus and distinguished from the drying-out form at the end of the 19th century [82]. In 1922, the term Esca was introduced to designate this disease characterized by the presence of white rot and the apoplectic form [80]. Since the 1990s, research on the Esca disease has increased along with the increase in the number of infested vineyards in several European countries such as Germany, Italy, and Greece [83]. This disease is mainly present in European countries (Germany [84], Austria [43], Bulgaria [85], Croatia [86], Spain [87], France [88], Greece [89], Hungary [90], Italy [91], Montenegro [92], Portugal [93], Romania [94], Serbia [95], Slovakia [96], Slovenia [97], Switzerland [98], Czech Republic [99], and Ukraine [100]). It is also ubiquitous in America (USA [101], Canada [102], and Mexico [103]), Asia (Iran [104], Israël [105], and Turkey [106]), Africa (South Africa [107], Algeria [34], and Morocco [108]), and Australia [13] (Figure 1).

Esca disease groups several syndromes caused mainly by vascular fungi that can invade the vine plants, either during injuries or during handling in the nursery. At present, numerous studies have shown that three fungal species are associated with Esca. *Phaeomoniella chlamydospora*, *Phaeoacremonium minimum*, and *Fomitiporia mediterranea* are a few examples [109]. Several more trunk pathogenic fungi have been isolated from vines exhibiting Esca symptoms in addition to these three, but their role and interaction with other fungi have yet to be elucidated [110].

*Phaeomoniella chlamydospora*: *P. chlamydospora* is one of the pioneers of Esca syndrome [64]. This anamorphic pathogen is also associated with Petri dish disease, which is reported in South Africa, Europe, the USA, and Australia [111]. It belongs to the Ascomycetes class of the *Herpotrichiaceae* family [112]. Its sexual form is unknown. The fungus is airborne. It penetrates through pruning wounds during the winter period. Its propagation can also take place during the vegetative period of the vine [113]. Moreover, *P. chlamydospora* spreads through rootstocks and grafts in nurseries. It has also been pointed out that green operations such as disbudding, leaf removal, thinning, and pruning are undeniable ways for the fungi to penetrate [64]. *P. chlamydospora* is classified as a blue stain fungus because it does not degrade cell walls [114]. The enzymes produced and identified are polygalacturonases, polymethylgalacturonases [115], β-glucosidases, and endo-β 1,4 glucanases. Extracellular polysaccharides have also been identified [116].

*Phaeoacremonium minimum*: *P. minimum*, named previously *P. aleophilum*, is one of the predominant *Phaeoacremonium* species found on vines [79]. This species is described as an intermediate genus between *Phialophora* and *Acremonium* [117]. More than 25 species of *Phaeoacremonium* genus have been isolated from vines affected by Esca [118]. These include *P. aleophilum*, *P. angustius*, *P. inflatipes*, *P. mortoniae*, *P. rubrigenum*, *P. viticola*, and *P. parasiticum*. It should be noted that *P. aleophilum* is generally present in an asexual form in vineyards. Sexual forms of different species of *Phaeoacremonium* have been found on the vine. This is the case for the example of *P. minimum* [19]. This fungus spreads by air and contaminates pruning wounds during the vegetative period. The excoriated trunk and arms are the source of inoculum [64]. According to Gubler et al. (2004), *P. minimum* is classified as a soft rot fungus because it grows inside the secondary walls forming cavities [119]. Its enzymatic background consists mainly of xylanases, β-glucosidases, and endo-β 1,4 glucanases, as well as seven other metabolites identified as potential toxins (naphthalenones).

*Fomitiporia mediterranea*: *F. mediterranea* is a basidiomycete of the *Russulales* order and *Hymenochaetacae* family. This pathogen is involved in the Esca syndrome [120]. It is classified as a white-rot fungus because it completely degrades cell walls. The biological life cycle of this fungus has remained elusive. *F. mediterranea* only propagates if temperatures are above 10 °C and the hygrometry rate exceeds 80% [121]. In addition, it is conducted with an arsenal of enzymes, namely xylanases, cellulases, β-1,3 glucanases, laccases, peroxidases, and phenoloxidases. Other phydroxybenzaldehyde-like metabolites are produced by the fungus [122].

*F. mediterranea*, *P. minimum*, and *P. chlamydospora* penetrate inside the trunk and twigs of the vine through moist and recent pruning wounds. This is accomplished by the basidiospores that germinate on the surface of the wounds, or through the mycelial fragments that can be carried by the pruning instruments. Inside the vine, the fungi disorganize the cells. This disorder can descend all along the vine. The fungi destroy the wood by killing first the infected parts, following the development of the mycelium inside the deadwood. The fungi may produce potential ligninolytic enzymes capable of degrading the lining inside [123]. The wood becomes soft and friable, forming a spongy and yellowish tissue [124]. According to Larignon et al. [9], the fungi *Phaeoacremonium* spp. and *P. chlamydospora* serve as pioneers of the Esca that can attack, digest, and transform the wood into tinder [9]. By contrast, *E. lata* causes hard brown necrosis in the lower part of the trunk [44] (Figure 3).

Despite past research efforts, the information regarding the disease epidemiology remains unknown. In 2000, a survey was carried out in Tuscany, Italy to study the epidemiology of Esca. The findings of the survey confirmed that in all the studied vineyards, disease incidence varied from year to year, perhaps about the amount of summer rainfall and/or the air temperature. The data revealed that it is very difficult to predict the occurrence of external Esca symptoms in a given year. Infected plants operate in a way that seems completely erratic. Plants that are strongly symptomatic one year may grow and produce almost normally the next, and plants that outwardly seem normal in one year (although the wood of the trunk is always more or less rotted) will not vegetate at all the following year [125].

The external symptoms of Esca can appear in different forms, slow and lightning or apoplectic forms of the herbaceous organs at the beginning of the summer period [9]. This expression varies from one year to another.

*The slow form*: This pathology, characterized by very specific functional signs, is the most frequent form. The disease is accompanied by the appearance of yellow spots on the limb, thus causing discolorations on the interveinal area and the edges of the leaves. Reddish colors appear in the centers of these discolorations, develop and then fuse, therefore creating larger patches. The red leaf blade is then separated from the green veins by a light yellow border. The leaves show, therefore, a characteristic tiger-like appearance [9]. Symptoms onset is preceded by a decrease in photosynthetic activity, a change in oxidative metabolism, and the expression of defense genes in the leaves [126]. Accordingly, the leaves eventually perish and fall off. This symptomatic manifestation can affect either the whole plant, a single arm, or a few twigs (Figure 4a). The manifestation of symptoms is particularly observed during mild and rainy summers. It is important to mention that the leaves from the lower part of the twigs are affected first [127,128].

*The apoplectic form*: Usually occurring in dry climates and in midsummer, the apoplectic form is most often similar to lightning or apoplectic appearance and is much more spectacular in the vineyard. Within a few days, there is a sudden loss of leaf turgidity, resulting in a greyish-green coloration that indicates the wilting of the entire plant or a branch [129]. This lightning form is characterized by a drop in the photosynthetic activity of the leaves one week before its expression [130]. A cross-section shows two types of necrosis characteristic of the Esca differing in their position and the number of zones in which they colonized [113]. The first is clear and soft necrosis in a central position. This necrosis, which contains several zones, is characterized by a black border surrounding a white rot and is separated from healthy wood by a pinkish-brown zone. At the periphery of the necrosis, black punctuations are regularly observed. Brown necrosis can also be noted in the central position consisting of blackish and pinkish-brown zones. This necrosis, also called pre-Esca necrosis, always precedes the light and tender necrosis in the central position. As the second type of necrosis, it is clear and tender in a sectorial position. It contains two zones: a chamois-brown zone, limiting the white rot. This necrosis is preceded by another necrosis of a brownish buff color located in a sectorial position, characterizing the eutypiosis disease [131] (Figure 4b).

### 2.3. Botryosphaeria Dieback

*Botryosphaeria dieback*, previously named Black Dead Arm (BDA), has been known as slow apoplexy. The term BDA was coined by Lehoczky in 1988 to make a distinction between this disease and the “Dead Arm Disease” caused by *Phomopsis viticola* [132]. The name comes from the fact that the phloem and xylem tissues of infected woody areas show a black coloration [133]. This disease attacks the framework of the strain, causing long-term death. This dieback was often attributed to Esca, given the similarity of symptoms in the vegetation [134].

The geographical distribution of this pathology is often linked and confused with the slow form of Esca disease. It is described in Chile [135], Spain [136], France [9], Hungary [132], Iran [137], Italy [138], Lebanon [139], Portugal [140], and Turkey [141] (Figure 1).

Very little is currently known about the epidemiology of *Botryosphaeria* diseases of grapevines. The enormous and increasing number of *Botryosphaeria* species found on grapevines makes epidemiological research of this pathogen more difficult. Species of *Botryosphaeriaceae* can differ in their epidemiology, the disease symptoms they cause, and their relative importance. Úrbez-Torres highlighted the importance of understanding the epidemiology of *Botryosphaeriaceae* species by studying sources of inoculum, conditions that endorse spore release, seasonal spore release patterns, seasonal susceptibility of pruning wounds, and factors that favor infection [142].

According to research, varied species require different climatic conditions to create fruiting structures. As a result, *Botryosphaeria* infection could occur in a variety of climatic conditions [143]. Moreover, the longer the period of wetness and high relative humidity extends, the more spores are produced and released, hence creating a much higher inoculum load and increasing the severity of infection [144]. According to several authors, an increase in wetness duration combined with high inoculum levels led to an increase in severe pistachio and peach tree infections [145,146].

The period of the pruning woods being susceptible to infection has not been thoroughly investigated. In the past, studies have shown that wounds are susceptible to infection for up to 4 months after pruning [20], but that susceptibility diminishes as the interval increases between pruning and infection [142]. *Botyosphaeriaceae* species have been found overwintering on dormant canes, in diseased wood, and on pruning debris on the vineyard floor mainly as pycnidia [132,147], which has been recognized as one of the most important inoculum sources in the field [142].

*Botryosphaeriaceae* species develop over many years in a vineyard, leading to a general loss in vigor and productivity of the grapevines, with symptoms rarely seen in one-year-old canes. Nonetheless, Larignon and Dubos (2001) discovered dark lesions on 1-year-old canes that were artificially inoculated with two *Botryosphaeriaceae* species [148]. In addition, *Botryosphaeriaceae* species have also been isolated from propagation material (young vines), rootstock mother vines, and failed graft unions of young plants in nurseries [149,150]. A recent study has reported that conidia of two species of *Botryosphaeriaceae* can be dispersed up to 2 m from the inoculum point in a single rainfall event [151].

In the herbaceous part, the first symptoms appear early and are present throughout the vegetative period. They can appear on one or more parts of the plant. The leaves of the lower part of the plant are most often affected first. Symptoms can evolve very rapidly (severe form) or pass through different phases (slow form), leading to premature leaf fall (Figure 5a). The slow form results in delayed grape ripening and a thinning of the foliage, which can give a tiger-like appearance to the leaves. Consequently, a set of reddish (black grape varieties) or yellow (white grape varieties) spots can be developed. The severe form is characterized by a drying of the inflorescence or clusters or rapid defoliation of the branches [14,152,153,154].

As for the wood, the bark peeling shows a brown strip a few centimeters wide, which starts from the twig and reaches the weld or even the rootstock. A cross-section made in the wood shows, at the edge of the brown band, a yellow to orange zone, limited to a few millimeters deep, in which the vessels are obstructed. These brown bands are most often in association with sectorial necroses, which are different from those of Eutypa dieback, not only by their color but also by their texture. The color is rather gray for the canker observed in plants affected by *Botryosphaeriacae*, and buff-brown for those affected by Eutypa dieback [155] (Figure 5b,c).

In the *Botryosphaeriaceae*, 21 different species were declared presently associated with Botryosphaeria dieback, the most frequent species isolated from these different necroses are *Diplodia seriata*, *Diplodia mutila*, and *Neofusicoccum parvum*. They are found on different plants and can cause a large number of dieback events [128,156]. Little information is given on the type of decay they cause in wood and on their enzymatic equipment. *D. seriata* secretes tyrosol, melleins, mellein derivatives, and 4-hydroxybenzaldehyde [157,158,159]. However, no relationship between pathogenicity and the level of toxins produced has been demonstrated. The life cycle of fungi on the grapevine is poorly known. Sporulation takes place during the vegetative period and seems to be independent of rainfall [160]. The way of entry of these pathogens is uncertain; however, infection via pruning wounds is strongly suspected [9]. They are preserved in the form of pycnidia that are localized either on the vine (trunk, arms, and pruning wounds) or pruning left on the ground [64].

## 3. Identification of Pathogens

### 3.1. Diagnosis of the Disease and Molecular Identification

In the absence of curative treatment against GTD, one of the most effective strategies to prevent its spread is to be able to early detect infected vines. Several research studies are thus focused on the development of diagnostic methods [8].

The identification of fungal species has undergone a spectacular evolution over time, from classical identification to identification via different tools such as conventional polymerase chain reaction (PCR), random amplified polymorphic DNA (RAPD), restriction fragment length polymorphism (RFLP), etc. In the case of grapevine decline, different studies have been conducted for PCR diagnosis either directly with DNA extracted from the necrotic wood or indirectly with DNA extracted from the mycelium re-cultured from the necrotic wood fragment [77,161,162,163].

Conventional PCR is a tool that targets in vitro replication techniques to obtain large quantities of a specific DNA fragment of defined length [134]. This is achieved by targeting specific regions in the fungal genome. The majority of wood disease diagnostics by PCR target internal transcribed spacer (ITS) sequences of fungal ribosomal ribonucleic acid (rDNA), or other gene sequences encoding for actin or β-tubulin [19,164]. The RAPD technique help construct genetic maps [165], as the generated PCR products from RAPD give a unique profile on the gel, making it possible to identify the desired fungus by comparing the profiles of the studied fungi to the profiles of other reference fungi. This amplification allowed the development of primers specific to amplified Sequence characterized amplified region (SCAR). However, in the case of GTD, several primers were used for the identification of *F. mediterranea* and *P. chlamydospora* associated with Esca disease [166], *E. lata* responsible for Eutypa dieback [162], and *D. seriata* linked to the *Botryosphaeria* dieback [167] (Table 1).

### 3.2. Real-Time PCR

Real-time PCR platforms are at the cutting edge of diagnostics, combining PCR chemistry with fluorescent probes/dyes to detect amplicons in a single closed reaction, reducing the risk of contamination. Fluorescence is generated in proportion to the amount of amplified DNA, which provides quantitative results in “real-time” without post-amplification analysis. Major detection formats include fluorescence resonance energy transfer (FRET) probes, TaqMan probes, dual probes, fusion curve analysis, and molecular beacons [172,173,174]. Billones-Baaijens et al. [171] have developed qPCR primers that can detect and quantify multiple species of Botryosphaeriaceae from the environment. It is the first qPCR method developed to target spores of multiple species. The developed protocol in this study was able to distinguish 10 species. Furthermore, the quantification was not affected by the nontarget DNA present in the samples [171].

Pouzoulet et al. [175] have designed two sets of qPCR primers that can target the beta-tubulin gene of two fungi (*D. seriata* and *E. lata*) for their use in qPCR SYBR Green chemistry. Moreover, the authors were able to detect and quantify the two fungi from naturally and experimentally infected samples during different conditions. These analytical approaches can improve and give a full idea about the etiology of both Eutypia and Botryosphaeria diebacks on the grapevine, which will help in disease management. Additionally, the study demonstrated the potential of this test to detect and track both fungi in wood samples from the field and prove the high sensitivity and accuracy of the detection to the traditional microbiological method.

## 4. Grapevine Defense Mechanisms against Fungal Attack

When a plant is threatened, it activates a series of genes that code for various effectors, receptors, signaling molecules, and protective molecules. Recognizing the genetic and molecular basis and identifying important genes involved in defense may provide valuable insights for GTD management. Grapevines, like other plants, attempt to fight back by activating defensive mechanisms namely the antioxidant system, the phenylpropanoid pathway, pathogenesis-related (PRs)-proteins, and phytoalexin synthesis, among others [176]. PRs proteins are an essential component of innate immune responses under biotic and abiotic stress. The PRs proteins protect the plants from infection by accumulating in non-infected tissues as well as in damaged and adjacent structures. The hypersensitive response (HR) and systemic acquired resistance (SAR) to infection are likewise mediated by PRs proteins. In response to any stress scenario and/or invading pathogen, the creation and activation of PRs proteins are essential. During incompatible host–pathogen interactions, the plant’s defensive responses restrict the damage caused by the pathogen.

Since the discovery of PRs proteins, the regulation of PR gene expression has been a highly active research area. The signaling that promotes pathogen-induced PR gene expression in plants, on the other hand, is yet unknown. This is due partly to the variety of environmental stimuli and phytohormone stimulation, which can cause the expression of several PRs genes [177]. PRs were discovered not for their anti-pathogenic properties, but rather for their ease of detection in infected plants. PRs are induced in a variety of plant species from various families, implying that these proteins play a broad role in biotic and abiotic stress adaptation. Many plant species from many families are induced in PRs, suggesting that PRs have a broad protective effect against biotic stress [178].

Nevertheless, the host’s arsenal of protection against aggressors is frequently ineffective to stop the disease from spreading depending on the pathogen’s lifestyle (necrotroph, biotroph, or hemibiotroph).

Different hormone-mediated signaling pathways regulate transcriptional reprogramming and, more importantly, plant defense mechanisms [179]. Based on *Arabidopsis thaliana* studies, Jasmonic acid (JA) and ethylene (ET) mediate defense responses to necrotrophic pathogen defense, which is boosted locally and systemically when microorganisms release cell wall-degrading lytic enzymes [180]. In addition to wall components, some phospholipids released by plasma membrane degradation directly trigger JA biosynthesis [181]. The increased expression of defense genes such as glucanases, chitinases, protease inhibitors, and enzymes involved in the formation of secondary metabolites such as phytoalexins, is triggered by the increase in JA levels [182]. Salicylic acid (SA), on the other hand, is essential for resistance to biotrophs and hemibiotrophs, as it leads to an increase in reactive oxygen species (ROS) and, as a result, localized programmed cell death (PCD) in infected tissue [183]. This HR restricts pathogen growth by limiting their access to nutrients and water [182]. Plants must be able to recognize these pathogens for these events to occur. During attacks, the signaling molecules SA [184,185] and ROS [186], upregulate acidic PRs, whereas basic PRs are upregulated by gaseous phytohormone ethylene and methyl jasmonate [187].

Currently, three types of plant–pathogen interactions have been documented. The first, PTI or PAMPs-triggered immunity (recently renamed MTI or MAMPs-triggered immunity), is a non-adaptive pathogen-triggered immune response. This corresponds to the first line of defense, common in plants of the same species facing pathogenic microorganisms [183]. PTI is mediated by plasma membrane-localized pattern recognition receptors (PRRs), which have an extracellular domain able to detect PAMPs and an intracellular domain that amplifies the signal inside the cell [188]. The second form of interaction is the effector-triggered susceptibility (ETS), termed after the capacity of certain microorganisms to overcome the baseline plant response by secreting virulence factors (effectors) that inhibit PTI, hence promoting the disease [183]. Lastly, effector-triggered immunity is a third sort of interaction (ETI). Receptors known as resistance proteins (R) allow plants of a specific genotype to recognize pathogen effectors. If an effector is identified by an R protein, either directly or indirectly, it is classified as an avirulence factor (AVR) and then the pathogen becomes avirulent to that plant since this interaction promotes the activation of HR [189,190].

The PR proteins have been classified into 17 families based on molecular mass, isoelectric point, localization, and biological activity, including β-1,3-glucanases, chitinases, thaumatin-like proteins, peroxidases, ribosome-inactivating proteins, thionins, nonspecific lipid transfer proteins, oxalate oxidase, and oxalate oxidase-like proteins [191]. Chitinases and β-1,3-glucanases are two essential hydrolytic enzymes that accumulate in a variety of plant species after infection by various pathogens. Chitinases, for instance, are extensively dispersed across the kingdoms of plants, animals, fungus, and bacteria. These enzymes catalyze the cleavage of a bond between C1 and C4 of two consecutive N-acetyl-D-glucosamine monomers of chitin, which is found in fungal cell walls and arthropod shells. Plant chitinases are usually endo-chitinases that may degrade chitin as well as suppress fungal growth. Many studies have found that chitinases, together with β-1,3 glucanases, play a significant role in plant defense against fungal infections. Various investigations in the sugar beet [192], wheat [193], and tomato [193] demonstrated that chitinase expression is upregulated by phytopathogen systems, and resistant types have higher upregulation than susceptible varieties. Transformation of chitinase genes was performed in tobacco [194], grapevine [195], rice [196], and peanut [197], achieving, thus, enhanced disease resistance. Plant β-1,3-glucanases, on the other side, belong to the PR-2 family of pathogenesis-related proteins and reportedly play a key role in plant defense responses to pathogen infection. Plants, yeasts, actinomycetes, bacteria, fungus, insects, and fish have all been reported to have these enzymes [198]. These enzymes catalyze the cleavage of the β-1,3-glucosidic bonds in β-1,3-glucan, another major structural component of the cell walls of many pathogenic fungi [199]. It has been proposed that β-1,3-glucanases hydrolyze fungal cell walls, causing the lysis of fungal cells when defending against fungi. β-1,3-glucanases also cause the synthesis of oligosaccharide elicitors in response to pathogen encounters, which elicit the production of other PR proteins or low molecular weight antifungal compounds, such as phytoalexins [200]. For example, Camps et al. [201] reported an up-regulation of several genes encoding PR proteins (thaumatin and osmotin, chitinase, and β-1,3-glucanase) in leaves of infected rooted cuttings (Carbernet Sauvignon) artificially infected with *E. lata* [201]. The work of Mutawila et al. [202] showed that the elicitation of cell suspension culture of *V. vinifera* cv Dauphine with *E. lata* culture filtrate resulted in an induction of VvPR2 (β-1,3-glucanase), VvPR5 (thaumatin and osmotin-like proteins), VvPR3 and VvPR4 (chitinase), and VvPR6 (protease inhibitor, PIN). Secondary metabolite induction is frequently linked to both defensive and pathogenic reactions [202,203,204]. Secondary metabolism is highly induced after infection of grapevine by numerous pathogens, including *E. lata* [176].

Several reports have indicated an upregulation of PAL, which encodes the first enzyme of the phenylpropanoid pathway, as well as genes coding for enzymes of the flavonoid and stilbenoid pathways, chalcone synthase (CHS), and stilbene synthase (STS), respectively [202]. Grapevine phytoalexin is a stilbene compound including the 3,5,4c- trihydroxystilbene or resveratrol and derivatives [205]. The phenylalanine/polymalonate pathway is used to synthesize stilbenes. The enzymes PAL and STS are required for resveratrol synthesis. Both genes were found to have coordinated expression in *V. vinifera* cv. Optima cells were treated with a fungal cell wall preparation [206] and leaves of *Cissus antarctica* were treated with UV light [207]. Grapevine resistance is enhanced by stilbenes. They are found in large concentrations in the heartwood of grapevine trunks, where they effectively inhibit wood decay caused by fungi. They are also collected in leaves or berries in response to infection by pathogens such as *P. viticola* or *B. cinerea*, and their antifungal effects have been investigated. Resveratrol has low antimicrobial action [208], but it is a precursor to more active derivatives and is accumulated in high concentrations in response to elicitation or pathogen attack. There have been established correlations between the ability of grapevine varieties or species to produce stilbenes and their resistance to cryptogamic diseases. Additionally, foreign expression of the stilbene synthase gene in plants or overexpression of the gene in grapevine usually results in increased pathogen resistance. Resveratrol inhibits *E. lata* mycelium growth in vitro [209], but it is unclear whether resveratrol and other phenolic compounds inhibit *E. lata* wood colonization in vivo [202].

According to recent data about grapevine/*E. lata* interaction, grapevine exhibits some of the typical responses of the PTI, such as PR-protein synthesis and secondary metabolites accumulation, suggesting that this fungus is sensed by the host [210]. Most *V. vinifera* cultivars, on the other hand, are vulnerable to *E. lata*, implying that defense responses are insufficient to prevent infection [202]. In grapevine, several PR proteins are synthesized upon infection through recognizing MAMPs (microbe associated molecular patterns) or DAMPs (damage-associated molecular patterns) such as oligosaccharide, lipid, and proteinaceous elicitors [211]. The majority of PR-proteins have direct antibacterial properties, (e.g., osmotin and thaumatin) via hydrolytic activity on pathogen cell walls, (e.g., glucanase and chitinase) and/or indirectly lead to the production of elicitors that trigger additional defense responses [212]. Following infection with a wide range of pathogens such as *Botrytis cinerea*, *P. viticola,* and *E. necator*, numerous studies have revealed the selective expression of PR-protein producing genes in distinct grapevine cultivars [176]. Up to now, few defense genes are known in grapevine, mainly PAL, VST or STS (stilbene synthase), LOX (lipoxygenase), CHIT (chitinase), GLU (glucanases), and PGIP (polygalacturonase inhibiting protein) and their expression have been studied in response to various elicitors. The recent sequencing of the grapevine genome now allows researchers to investigate variations in global gene expression in response to elicitors or pathogen infection [213].

All of these findings showed that resveratrol, derivative chemicals, and flavonoids play a role in cell wall strengthening in response to infection or elicitor treatment. Numerous chitinases (PR3, PR4, PR8, and PR11) have been found in leaves that are either constitutive or inducible by wounding, SA treatment, or infection with *P. viticola*, *E. necator*, or *B. cinerea* [214]. Osmotins or thaumatin-like proteins [215], a ribonuclease-like protein [216], and a lipid transfer protein or LTP (PR14) have all been identified as PR proteins. These PR proteins may be involved in grape defense, but this hypothesis remains to be ascertained. The grapevine phytoalexins resveratrol and its derivatives have been extensively researched. In addition to antibacterial activity, they may promote cell wall strengthening. Elicitors’ mode of action and activity is controlled by their chemical structure. Elicitors seem to be of special relevance for crop protection since they cannot only elicit defenses in a wide range of plants, but they are also usually non-toxic and appropriate for industrial manufacturing from readily available sources. Despite encouraging results, the use of induced resistance in the vineyard is still plagued by inconsistency and yet has only provided minimal disease control.

## 5. Molecular Mechanisms of Fungal Endophytes Colonization

Several associations were reported between fungi and plants such as mycorrhizas, parasitism, myco-heterotrophic, and antagonism. These associations can benefit or be detrimental and lead to the death of the plant. Much attention is being paid to the molecular mechanisms underlying fungal endophytes colonization, whilst only a few studies were carried out to examine the mechanisms/interactions between plant and fungal endophytes [217,218].

To colonize the plant, the endophytic fungi need to break the cell walls which is the first physical wall to go through. The different molecular exchanges between the host and the endophytic fungi took place through the plasma membrane and the cell wall. Later, this is deeply involved at the molecular level by mediating most interactions.

Endophytic fungi secrete cell wall degradative enzymes such as pectinase, cellulase, laccase, etc., which promote the degradation of the cell wall and cause structural damage, helping in the colonization of plant tissues [219].

As an example, ericoid mycorrhiza has a similar degrading enzyme gene content library as pathogens and saprophytic fungi, like the polysaccharide degrading enzymes, lipases, proteases, and some enzymes involved in secondary metabolism. The genome of ericoid mycorrhiza contains numerous numbers of the CAZymes, quinone-dependent oxidoreductases, and iron reductase, then other endo and ectomycorrhizal fungi genomes. Their genome encodes for several genes coding and enzymes such as cellobiose dehydrogenases, lytic polysaccharide monooxygenases, as well as laccases which are involved in the cleavage of different compounds such as cellulose, chitin, pectin, and hemicellulose [219].

As an example of a biotrophic fungus, the *Phytophthora infestans* (Mont.) is a destructive pathogen to plants; it uses both cytoplasmic and apoplastic effectors during host colonization. These effectors contain RXLR (conserved motifs -arginine, any amino acid, leucine, and arginine) and are potentially involved in transcriptional regulation in the host. The pathogen uses AVR3a, a cytoplasmic RXLR effector which helps in the stabilization of U-box that contain E3 ligase protein (CMPG1) during the infection of the plant [220]. The degradation of this ligase protein leads to the induction of a form of a plant cell death named infestin 1 (INF1)-triggered cell death (ICD). The RXLR effector, i.e., AVRblb2 enhances fungus pathogenicity by inhibiting the release of a papain-like cysteine protease (C14), which plays a role in the inhibition of the fungus infection in the host. These effectors have shown a dual function. They not only invade the host physiology of susceptible plants but also affect the immune response of resistant plants [220].

## 6. Control Methods

So far, no grape varieties are known to be immune to GTD. Since the banning of sodium arsenite because of its toxicity, a worrying progression of diseases in vineyards around the world is taking place. Only Esquive^®^ produced by Agrauxine, and Vintec^®^ produced by BELCHIM Crop Protection, have been authorized to fight against these diseases. Other products are being studied, including molecules capable of transporting the pathogen through the plant [221,222,223].

The major problem with GTD lies in the total absence of curative treatment, as the pathogen spreads easily. The application of prophylactic measures is recommended, although their effectiveness is discussed. Good prophylaxis involves the systematic and total elimination of all deadwood, whether from the vine or a surrounding forest [153,224]. Storage of pruning wood close to vineyards is prohibited, and it is preferable to remove this wood quickly to avoid inoculum maintenance. Late pruning of susceptible grape varieties can also be considered to have less receptive wounds.

According to prior studies, more than 90 active ingredients were investigated from 2000 to 2018 against the Esca diseases, Botryosphaeria dieback, and Eutypa dieback. The majority of the active substances tested are synthetic organic compounds that are employed singly or in mixtures. The most commonly tested chemical groups include benzimidazoles, triazoles, and strobilurins [224].

The most efficient benzimidazoles were benomyl, carbendazim, and thiophanate-methyl, which consistently showed great efficacy in both the lab and in the field for pruning wound protection and nurseries, regardless of the geographical area or GTD targeted. Their effectiveness against the major GTD was mostly attributable to their broad-spectrum fungicidal action, persistence, and systemic activity. Except for *E. lata*, they demonstrated high competence in decreasing both mycelial growth and conidial germination in vitro testing. Benzimidazoles were used to protect vines against new infection in pruning wounds, mostly as preventative treatments and, to a limited extent, as curative ones. A previous study reported a 14-day long-lasting preventive effect of benomyl in protecting wounds from *E. lata* infections, as well as a moderate curative effect of benomyl and carbendazim, unless the active ingredients were applied 1 day after *E. lata* inoculation [225]. Moreover, benzimidazoles sprayed or painted on pruning wounds showed similar efficiency. A disadvantage of benzimidazoles is that fungi can acquire resistance to them, as demonstrated by *P. minimum*’s resistance towards carbendazim [226]. In nurseries, benzimidazoles effectively reduced the presence of vascular Esca complex pathogens. When applied during hydration or before grafting, benomyl and carbendazim, for example, decreased the amount of pathogen inoculum in grafted plants [117,227].

Triazoles, on the other side, are the most common group among synthetic chemicals. They are currently used to prevent vineyards from a variety of diseases (powdery mildew, botrytis bunch rot, etc.). Triazoles have been tried to manage GTD for this reason, as well as for their systemic characteristics. In vitro tests demonstrated that it is very effective at inhibiting conidial germination and reducing the mycelial growth of GTD pathogens [228] Some of these positive in vitro effects were tested and verified *in planta* bio-assays.

Currently, strobilurins are used to control downy and powdery mildew in vineyards. They were examined to see if they could protect pruning wounds in particular. In both in vitro and wound protection trials, pyraclostrobin has shown to be the most effective against GTD pathogens. According to previous studies, other strobilurins were only investigated in vitro and exhibited various efficiency depending on the GTD pathogen [229].

The resistance of the pathogen to chemicals is one of the major constraints of phytosanitary treatments, it is due to the often-prolonged misuse of these products. Alternatives are at the center of farmers’ major concerns. Several biocontrol agents have been tested to limit wood diseases such as *Trichoderma*, which have a battery of potentially usable attack mechanisms. These fungi have antagonistic and hyperparasitic activity against a large number of microorganisms, particularly soil microorganisms, and are used in the biological control of various diseases on different crops [230].

It should be noted that various studies have explained the abundance of *Trichoderma* species in different ecosystems by their ability to produce various bioactive substances and enzymes that are responsible for the degradation of the cell walls of pathogens. They are therefore an important link in biological chains [231,232]. For their part, Gaigole et al. [233], consider *Trichoderma* as a cellulitic ascomycete.

The mycoparasitic activity of *Trichoderma* spp. against the sclerotia of plant pathogenic fungi is considered a powerful tool for efficient and effective biological control since these highly resistant vegetative structures represent the primary form of survival of the pathogen in the soil [234]. Singh et al. [235] have studied the interaction established between *Trichoderma* and the plant pathogenic fungi by histological and biochemical analysis and showed that several enzymatic activities are related to mycoparasitism, such as enzymes degrading cell wall components, namely, chitinase, cellulase, lipase, and protease or phenolic compounds, laccase degrading lignin and melanin.

The development of sophisticated biopolymer-based systems is the current trend in encapsulation application in agriculture, i.e., microcapsule formulations combining two active agents. Given the wide range of encapsulation procedures for encapsulated chemical agents, there are few investigations on simultaneous encapsulation and delivery of biological and chemical agents in the literature [236].

More recently, Peil et al. [237] have developed a spore-compatible layer-by-layer assembly to encapsulate the spores of a new mycoparasite strain of *Trichoderma reesei* IBWF 034-05 in a bio-sourced and biodegradable lignin shell creating a surfactant-free, self-stabilizing spore dispersion. The lignin shell protects the spores by transferring them into a dormant state. The spore dispersion that results is colloidally stable for several months and can be injected into the trunk. Once injected, encapsulated spores fulfill the role of a Trojan horse. If the plant is infected with Esca pathogens, the pathogens’ lignin-degrading enzymes degrade the lignin shell and initiate the germination process. *T. reesei* that has germinated is capable of parasitizing fungal infections and displacing them from their natural habitat. At the same time, *T. reesei* strengthens plants against subsequent infections. This concept allows *T. reesei* IBWF 034-05 to be used for both protective and curative treatments of Esca, one of the most infectious GTDs in the world [236,237].

Encapsulation prevents undesirable premature germination and allows application as an aqueous dispersion by injection into the trunk. The spores injected into the plant remain resting. When lignin-degrading fungi infect the plant, the shell is enzymatically degraded, and germination is selective. This concept allows *Trichoderma* spores to treat against the main GTDs such as Esca [237].

In addition, phytocompounds should be much more advantageous than synthetic pesticides since they are biodegradable, non-polluting, and do not have residual or phytotoxic properties. Ammad et al. [238] have demonstrated that the essential oil of lemon (*Citrus limon* L.) significantly inhibits the growth of three pathogenic fungi of vine wood (*Eutypa* sp., *B. dothidea*, and *F. mediterranea*). The findings revealed a novel usage of lemon essential oil to prevent fungal diseases of grapevine wood for the first time [238]. The antifungal potential of citrus essential oils has been widely highlighted [239,240]. Moreover, Van Hung et al. [241] reported that mycelial growth decreases with increasing essential oil concentration. Similar results were obtained by Mishra and Dubey (1994) [242], with essential oils of oranges (*Citrus Sinensis*) against *Aspergillus flavus* Link. On the other hand, the effect of lemon oil (*C. limon*) was observed against *Trichophyton mentagrophytes*, *Epidermophyton floccosum*, and *Microsporum gypseum* [243]. It seems that this citrus inhibitory effect is due to the richness of citrus essential oils in monoterpenes, as shown by Sawamura (2011) [244]. These are apolar compounds with great penetrating power [242]. They diffuse into fungal membrane structures and damage them by increasing their permeability. They also inhibit intercellular and extracellular enzymes and act as a regulator of cell metabolism by affecting enzyme synthesis in the nucleus or ribosome. Fisher and Phillips (2008) [121] highlighted that they interact with nutrients uptake from the environment, which affects mycelial growth in fungi.

Sanitary measures are also taken in nurseries. Indeed, these are major sources of contamination of future vines [245]. The hot water treatment of cuttings initially used to control “Flavescence dorée” and black wood is also being investigated as a low-cost alternative. Nevertheless, some grape varieties resist treatment poorly, resulting in losses [246,247]. Pierron et al. [248] studied a method of treating cuttings in nurseries with ozonated water to control Esca. This method, widely used in agriculture for post-harvest treatments or in food processing, has proven its antimicrobial properties. For example, it is effective against *B. cinerea* attacking table grapes [249] or against powdery mildew on cucumber leaves [250]. It is a highly oxidizing product with low persistence, causing spore germination to stop at 100% in vitro and 50% *in planta* [249].

## 7. Conclusions

The grapevine confronts a plethora of threats during its life, some of which act in the first period of its life (Petri disease, Blackfoot, Verticilliosis), and others later (*Eutypa dieback*, *Esca disease*, *Botryosphaeria dieback*). The several patterns of expression observed in the vineyard correspond to various disruptions in the plant’s metabolism in the presence of its aggressor.

Globally, the incidence of GTDs, mainly *Esca disease*, *Eutypa*, and *Botryosphaeria* diebacks, has risen dramatically in recent decades. In 1999, the International Council on Grapevine Trunk Disease (ICGTD) was created to facilitate the exchange of useful data on pathogen identification, detection, host–pathogen interaction, epidemiology, and disease management concerning GTD.

In the lack of precise information concerning the GTD, especially in Morocco, the present review is an update of scientific knowledge on GTDs, by targeting these fungal diseases in their entirety and diversity. To our knowledge, this is the first work focusing on these diseases in Morocco. Further studies are carried out to assess all the aspects related to the GTD complex.

## Figures and Tables

**Figure 1 jof-08-00595-f001:**
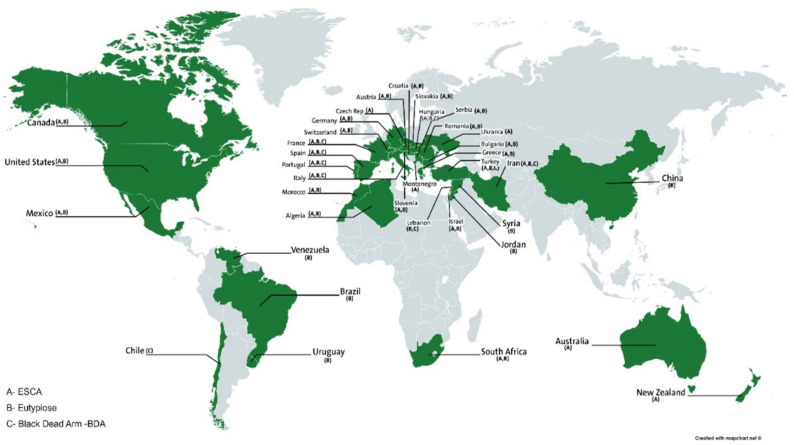
Global distribution of grapevine trunk diseases.

**Figure 2 jof-08-00595-f002:**
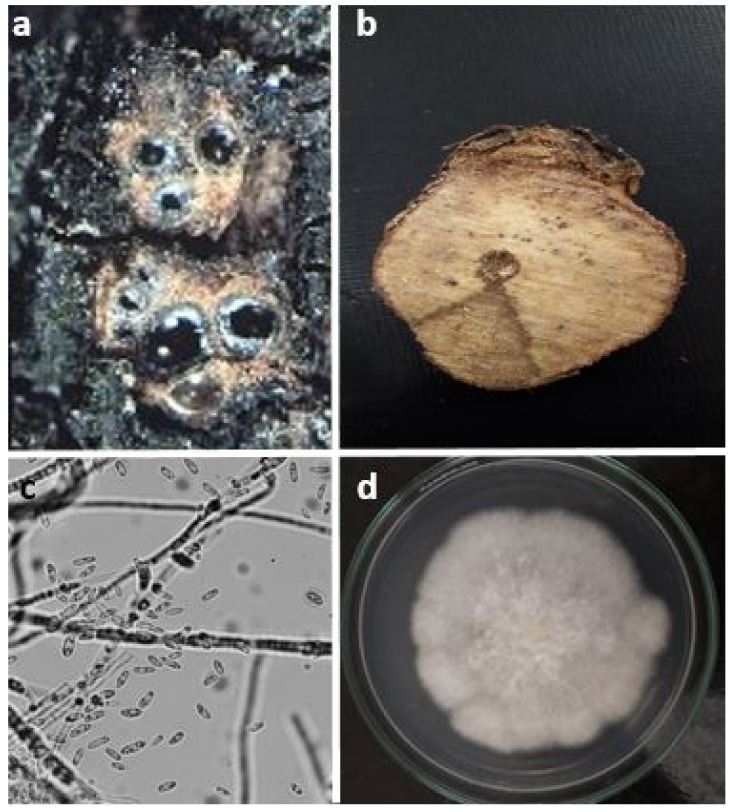
(**a**) Perithecia materializing the presence of the fungus on the arm of the vine; (**b**) typical symptoms of eutypiosis on the wood (V-shaped necrotic zone); (**c**) ascospores of *Eutypa lata*; (**d**) cottony white mycelium characteristic of *E. lata*.

**Figure 3 jof-08-00595-f003:**
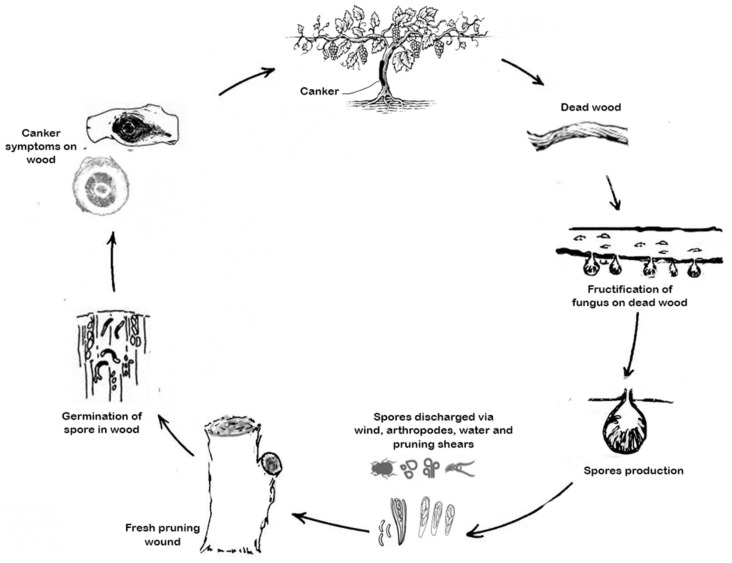
The life cycle of the fungus causing grapevine trunk disease.

**Figure 4 jof-08-00595-f004:**
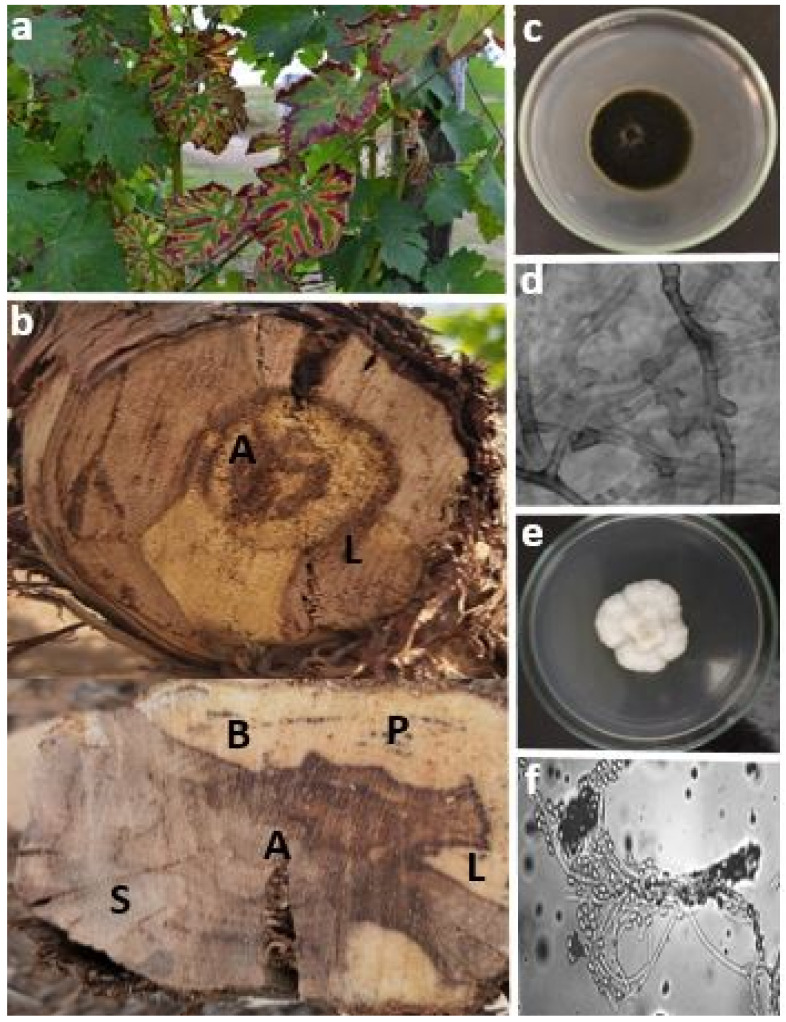
(**a**) Leaf symptoms of esca on grapevine; (**b**) necrosis observed in affected vines (**A.** white rot; **L.** interaction zone (border) located between white rot and healthy wood; **P.** black spots; **S**. sectorial necrosis; **B.** healthy wood); (**c**) morphological characteristics of *Phaeomoniella chlamydospora*; (**d**) globular chlamydospores formed within the mycelium; (**e**) morphological characteristic of *Phaeoacremonium minimum*; (**f**) conidiophore and hyaline conidia, oblong-ellipsoid to an allantoid characteristic of *P. minimum*.

**Figure 5 jof-08-00595-f005:**
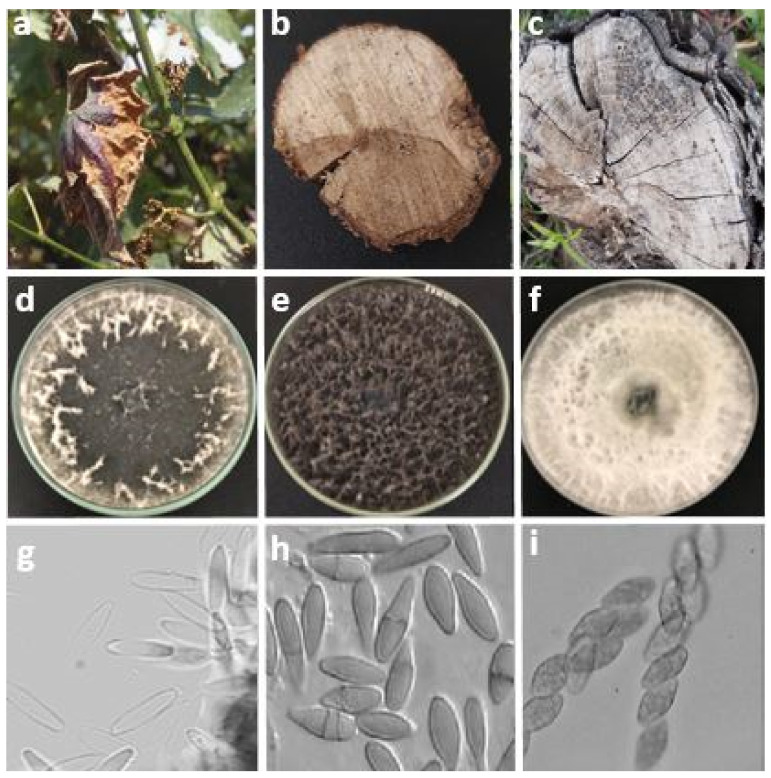
(**a**) Characteristic symptoms of Botryosphaeria dieback leading to leaf fall; (**b**) detail brown band and vessels obstructed by yellowish material; (**c**) grey sectorial necrosis; (**d**–**f**) appearance of the mycelia of *Botryosphaeria dothidea*, *Botryosphaeria stevensii* and *Neofusicoccum parvum* after 7 days incubation; (**g**–**i**) aspects of conidia of *Botryosphaeria dothidea*, *Botryosphaeria stevensii* and *Neofusicoccum parvum,* respectively.

**Table 1 jof-08-00595-t001:** Specific primers to detect different fungal species involved in vine wood diseases.

Fungus	Primer	Sequence	Length	Reference
*Eutypa lata*	Eut02 Eut02	5′TGGTGGACGGGTAGGGTTAG3′ 5′GGCCTTACCGAAATAGACCAA3′	643 bp	[162,168]
*Phaeoacremonium minimum*	PAL1 PAL2	5′-AGGTCGGGGGCCAAC-3′ 5′-AGGTGTAAACTACTGCGC-3′	415 bp	[169]
*Phaeomoniella chlamydospora*	Pch1 Pch2	5′-CTCCAACCCTTTGTTTATC-3′ 5′-TGAAAGTTGATATGGACCC-3′	360 bp	[169]
*Fomitiporia mediterranea*	Fmed1 Fmed2	5′-GCAGTAGTAATAATAACAATC-3′ 5′-GGTCAAAGGAGTCAAATGGT-3′	550 bp	[170]
*Botryosphaeriaceae* spp.	Bot-BtF1 Bot-BtR1	5′-GTATGGCAATCTTCTGAACG-3′ 5′-CAGTTGTTACCGGCRCCAGA-3′	410 bp	[171]

## Data Availability

The data used for the analyses in this study are available within the article.

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
