# Peer review of "A Panoramic View on Grapevine Trunk Diseases Threats: Case of Eutypa Dieback, Botryosphaeria Dieback, and Esca Disease"

_jof, 2022, doi:10.3390/jof8060595_

Round 1

Reviewer 1 Report

Remove the sentence:

32 the world's most widely cultivated fruit plant

Cannot find reference [3]

Do you mean US$? :

50 currency of about 10952902.00 $.

Reference [5] is a phd thesis. You should you write more information to locate it. 

Change from (1959-1936) to (1859-1936) 

155 to as folletage by Pierre Viala (1959-1936) 

Change from laccas to laccase

676 tinase, cellulase, laccas etc, helps in the degradation of the cell wall and causes damage 

Change from papainlike to papain-like 

695 hances fungus pathogenicity by inhibiting the release of a papainlike cysteine protease 

Author Response

  1. Remove the sentence: 32 the world's most widely cultivated fruit plant

 Answer: We agree with the reviewer’s comment, please check the revised MS.

  1. Cannot find reference [3]

Answer: We agree with the reviewer’s comment, please check the revised manuscript. 

  1. Do you mean US$? : 50 currency of about 10952902.00 $.

Answer: Yes. The currency used in this manuscript is the American dollar US$. Please check the revised MS.

  1. Reference [5] is a PhD thesis. You should you write more information to locate it. 

 Answer: We agree with the reviewer’s comment. New data were added to the manuscript.

  1. Change from (1959-1936) to (1859-1936)  :155 to as folletage by Pierre Viala (1959-1936) 
  2. Change from laccas to laccase :676 tinase, cellulase, laccas etc, helps in the degradation of the cell wall and causes damage 
  3. Change from papainlike to papain-like  : 695 hances fungus pathogenicity by inhibiting the release of a papainlike cysteine protease 

Answer: We agree with the reviewer’s comments. Please check the revised MS.

Reviewer 2 Report

Please revise your paper according to my comments in the attached file

Author Response

  1. 36-38 Add suitable reference to this sentence

Answer: We agree with the reviewer’s comment. The reference was added to the review. Please check the revised MS.

  1. 42-48 What are the sources of this data?

Answer: We agree with the reviewer’s comment. Please check the revised MS.

  1. 58- 59 Add some examples of common pathogens

Answer: We agree with the reviewer’s comment. Examples of the common pathogens were added to the MS.

  1. 65 Change this expression : to quickly but accurately

Answer: We agree with the reviewer’s comment. Please check the revised MS.

  1. 89 It is not clear : why did you mention few studies? Or do you mean there are few studies

Answer: We agree with the reviewer’s comment. To our knowledge; this is the first detailed study on grapevine trunk diseases in Morocco.

  1. 433 the isolation of fungi is commonly known and widely studied

Answer: We agree with the reviewer’s comment.

  1. 443-456 what are the importance for mention these morphological characteristics.

Answer: We agree with the reviewer’s comment. Since those data are already present in the literature. We deleted them from the manuscript. Please check the revised MS.

  1. 496 add number for this subtitle

Answer: We agree with the reviewer’s comment.

  1. 822- 841 Try to rewrite again the conclusion, it should be small and clear

Answer: We agree with the reviewer’s comment. Modifications were carried out. Please check the revised MS.

  1. 856 You add so much references, try to reduce this number

Answer: We agree with the reviewer’s comment. The number of references was reduced. Please check the revised MS.

Reviewer 3 Report

The review entitled "A Panoramic View on Grapevine Trunk Diseases Threats: Case 3 of Eutypa Dieback, Botryosphaeria Dieback, and Esca Disease" is scientifically important and suitable for publication.

However, it is important that grammer and paragraph structure are improve. For example, there are single sentence paragraphs (Lines 675-677). The authors also need to make sure that all sentences are clearly written and easy to understand. For example, in the following sentence "Several associations were reported between fungi and plants" the authors should rewrite the sentence so that it is clear what associations they are talking about (Line 666).

Author Response

The review entitled "A Panoramic View on Grapevine Trunk Diseases Threats: Case 3 of Eutypa Dieback, Botryosphaeria Dieback, and Esca Disease" is scientifically important and suitable for publication.

However, it is important that grammer and paragraph structure are improve. For example, there are single sentence paragraphs (Lines 675-677). The authors also need to make sure that all sentences are clearly written and easy to understand. For example, in the following sentence "Several associations were reported between fungi and plants" the authors should rewrite the sentence so that it is clear what associations they are talking about (Line 666).

Answer: We thank the reviewer for his valuable comments and all the modifications were carried out accordingly in the manuscript.